# Axonal T3 uptake and transport can trigger thyroid hormone signaling in the brain

**Federico Salas-Lucia[1†], Csaba Fekete[2†], Richárd Sinkó[3,4], Péter Egri[3], Kristóf Rada[3], Yvette Ruska[2], Balázs Gereben[3]\*, Antonio C Bianco[1]\***

[1]Section of Adult and Pediatric Endocrinology and Metabolism, University of Chicago, Chicago, United States; [2]Laboratory of Integrative Neuroendocrinology, Institute of Experimental Medicine, Budapest, Hungary; [3]Laboratory of Molecular Cell Metabolism, Institute of Experimental Medicine, Budapest, Hungary; [4]János Szentágothai PhD School of Neurosciences, Semmelweis University, Budapest, Hungary

**\*For correspondence:**
gereben.balazs@koki.hu (BG);
abianco1@uchicago.edu (ACB)

[†]These authors contributed equally to this work

**Abstract** The development of the brain, as well as mood and cognitive functions, are affected by thyroid hormone (TH) signaling. Neurons are the critical cellular target for TH action, with T3 regulating the expression of important neuronal gene sets. However, the steps involved in T3 signaling remain poorly known given that neurons express high levels of type 3 deiodinase (D3), which inactivates both T4 and T3. To investigate this mechanism, we used a compartmentalized microfluid device and identified a novel neuronal pathway of T3 transport and action that involves axonal T3 uptake into clathrin-dependent, endosomal/non-degradative lysosomes (NDLs). NDLs-containing T3 are retrogradely transported via microtubules, delivering T3 to the cell nucleus, and doubling the expression of a T3-responsive reporter gene. The NDLs also contain the monocarboxylate transporter 8 (Mct8) and D3, which transport and inactivate T3, respectively. Notwithstanding, T3 gets away from degradation because D3's active center is in the cytosol. Moreover, we used a unique mouse system to show that T3 implanted in specific brain areas can trigger selective signaling in distant locations, as far as the contralateral hemisphere. These findings provide a pathway for L-T3 to reach neurons and resolve the paradox of T3 signaling in the brain amid high D3 activity.

## Editor's evaluation

This novel study by Salas-Lucia examines retrograde transport of T3 in neurons using a compartmentalized microfluid device in-vitro and implantation of T3 crystals in the vivo models to understand the cellular mechanisms of T3 transport and activity in neurons. Furthermore, the authors show how T3 transport by this non-degradative lysosomal mechanism would activate genes in the nucleus. The experiments are well-designed and support the results and conclusions.

## Introduction

Thyroid hormones (TH) are crucial for brain development and influence brain function throughout life (*Ganguli et al., 1996*; *Joffe et al., 2013*; *Rovet, 1999*; *Bernal, 2017*; *Salas-Lucia et al., 2020*). However, the complex architectural organization of the brain and the unique properties of each cell type pose a challenge to the complete understanding of the mechanisms that regulate local TH signaling (*Diez et al., 2021*). Neurons are an important TH target in the brain as they express the highest levels of TH receptors (TRs) (*Crantz et al., 1982*). The blood-brain barrier (BBB) and glial cells

also play a role in TH signaling in the brain, regulating the amount of TH that reaches neurons through selective transport and metabolism (*Bernal et al., 2015*; *Morte and Bernal, 2014*). As a result, the human brain responds promptly to minor fluctuations in TH signaling by changing the expression of T3-responsive genes (*Marcelino et al., 2020*). TH transporters seem to play a critical role in T3 signaling as showcased by the profound brain hypothyroidism observed in boys carrying mutations in the monocarboxylate transporter 8 (MCT8, SlC16A2). The resulting Allen Herndon Dudley syndrome is marked by severe and irreversible neurological damage (*Dumitrescu et al., 2004*) due to reduced TH availability to neurons.

Plasma T3 can reach neurons via cellular transporters located in the BBB (*Wirth et al., 2009*; *Vancamp and Darras, 2018*), including MCT8—other species-specific transporters also play a role— but most T3 bound to TRs in the brain is originated from local D2 (*Crantz et al., 1982*; *Galton et al., 2007*), the enzyme that catalyzed conversion of T4 to T3. Within the brain, D2 is expressed in glial cells, astrocytes, and tanycytes, but not in neurons (*Guadaño-Ferraz et al., 1997*; *Tu et al., 1997*). Astrocytes are intimately related to neurons (hundreds of thousands of neuronal synapses per astrocyte); an array of metabolites (and even mitochondria) are known to be preferentially exchanged between astrocytes and neurons (*Weber and Barros, 2015*). Hence, the concept that the brain responsiveness to L-T4 is mediated by astrocyte-derived T3 production and transfer to neighboring neurons is well accepted (*Morte and Bernal, 2014*; *Freitas et al., 2010*). Accordingly, a mouse with glial-cell selective inactivation of the gene encoding D2 (Dio2) exhibits a mood and cognitive phenotype typical of hypothyroidism (*Bocco et al., 2016*).

Despite the local T4 deiodination and the intricate astrocyte-neuron T3 interplay, the brain also responds promptly to T3 administration (*Leonard et al., 1981*; *Salas-Lucia and Bianco, 2022*). Indeed, a replacement therapy containing LT3 was found to be superior for some patients with hypothyroidism (*Jonklaas et al., 2021*). In addition, short-term injections of L-T3 fully normalize the cognitive phenotype of the Ala92-Dio2 mouse, indicating that the impaired T4 to T3 conversion can be corrected with the administration of LT3 (*Jo et al., 2019*). The cerebral cortex is known for responding rapidly—within hours—to injections of L-T3 (*Leonard et al., 1981*), including induction of the Luc-mRNA levels in the THAI mouse. LT3 also promptly restores TH signaling in the brain of LT4-treated rats with hypothyroidism (*Werneck de Castro et al., 2015*). Nonetheless, that treatment with L-T3 can be effective is unexpected given that neurons express high levels of the TH inactivating type 3 deiodinase (Dio3) (*Tu et al., 1999*). Based on all we know about D3, its role in neurons should function as a barrier to incoming T3, minimizing or preventing T3 signaling (*Friesema et al., 2006*; *Hernandez et al., 2010*; *Martinez et al., 2022*). And yet, we know that brain TRs are nearly fully occupied with T3 (*Crantz et al., 1982*). Thus, it is not clear how T3 escapes from D3-mediated degradation in neurons and reaches the nucleus of these cells, and consequently how treatment with small amounts of L-T3 can be effective in restoring brain-localized hypothyroidism in mice and humans.

To solve this puzzle, here we studied the cellular mechanisms underlying the pathway through which T3 can enter and trigger biological effects in cortical and peripheral neurons. Our findings reveal that these neurons take up T3 in axonal termini via endosomal/non-degradative lysosomes and, after retrograde transport to the cell nucleus, initiate regulation of T3-responsive genes. These findings expand our understanding of T3 actions in the brain and have broad implications for L-T3-containing therapies for patients with hypothyroidism.

## Results

### Neuronal response to T3 requires MCT8 and retrograde TH transport

We first deconstructed the brain's architecture using a compartmentalized microfluidic chamber (MC), which purposely keeps the neuronal cell bodies and their long distal axons in two physically separated chambers. Primary cortical neurons (PCN) obtained from E16.5 mouse embryos were seeded in the cell side (MC-CS) and cultured for up to 13 days (*Figure 1—figure supplement 1A*). During this period, distal axons grew from parental cell bodies into the axonal side of the microchamber (MC-AS) (*Figure 1—figure supplement 1B*). By day in vitro (DIV) ~7, PCN had crossed the 450 μm long microgroove channels and by DIV ~10 densely populated the MC-AS (*Figure 1—figure supplement 1B*; *Figure 1A*). Adding calcein fluorescence to the MC-CS revealed labeled axons (1–3 axons/channels) reaching the MC-AS (inset in *Figure 1—figure supplement 1C*). The compartments remained

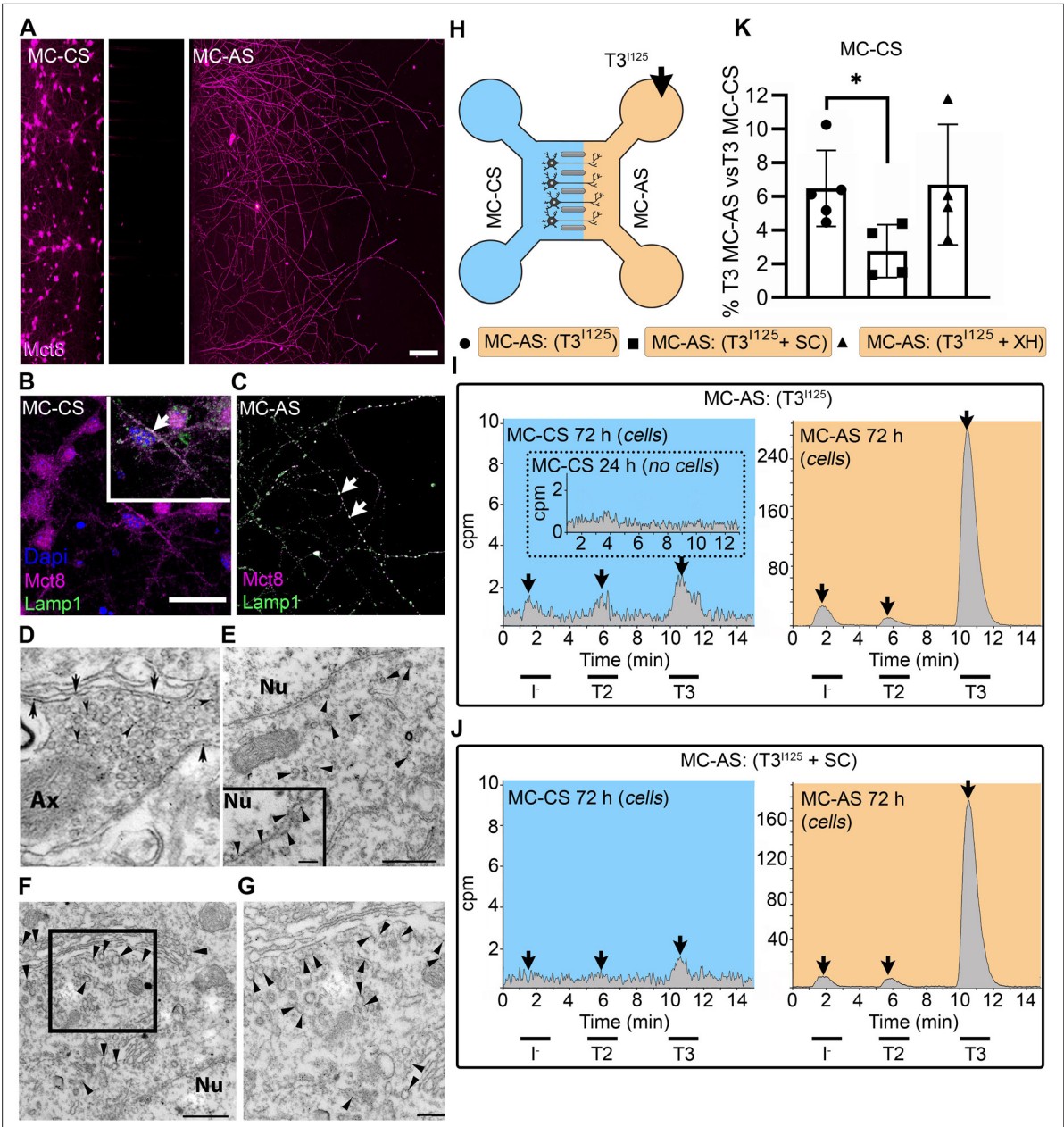

**Figure 1.** T3 is taken up and retrogradely transported through neuronal long distal axons. (**A-C**) Immunofluorescence of cortical neurons in the compartmentalized device using the indicated antibodies (**A**) Collage of a compartmentalized culture stained for MCT8 (magenta). (**B–C**) MCT8 staining in magenta, Lamp1 in green; colocalization is indicated as white. (**B**) In the MC-CS and (**C**) in the MC-AS. (**D**) MCT8-immunoreactivity (silver grains) was present in the outer cell membrane of neuronal elements, and in the membrane of vesicles of axonal profiles (arrows and arrowheads, respectively), on the nuclear membrane (inset in **E**), in vesicles close to the nucleus of the neurons (**E**), and the trans and cis Golgi apparatus (**F**). (**G**) Detail of the squared area in (**F, H**). T3$^{I125}$ was applied in the MC-AS. (**I**). After 72 hr, T3$^{I125}$ was detected in the MC-CS medium (no T3$^{I125}$ was detected after 24 hr; inset in the blue panel). (**J**). Effect of 2 µM SC on T3 uptake and transport. The size of the T3$^{I125}$ peak in the MC-CS decreased after 72 hr compared to (**I, K**). Quantitation of T3$^{I125}$ transported to the MC-CS medium under the indicated conditions. The Y-axis in % T3$^{I125}$ in the MC-CS medium vs. T3$^{I125}$ added to the MC-AS medium. Values are mean ± SD of 4–5 independent experiments; *p<0.05 in comparison with T3$^{I125}$ incubation. SC, Silychristin; XH, Xanthohumol; Ax, axon; Nu, nucleus. Scale bars are 25 µM on A-E and G, 150 µM on F, 500 nm on J, K, and M, and 200 nm on L.

The online version of this article includes the following source data and figure supplement(s) for figure 1:

**Figure supplement 1.** A system to study TH signaling in neurons.

**Figure supplement 2.** Neurons residing in the MC-CS are excitatory.

**Figure supplement 3.** T3 trafficking in rat DRG cells in microfluid chambers.

*Figure 1 continued on next page*

*Figure 1 continued*

**Figure supplement 3—source data 1.** Original blots for panels D and E.

**Figure supplement 4.** T3 is taken up and anterogradely transported through neuronal long distal axons.

fluidically isolated (*Figure 1—figure supplement 1D*). In the absence of cells, adding $T3^{I125}$ to the MC-AS revealed the predominant $T3^{I125}$ peak after 24–72 hr (minimal $T2^{I125}$ and $I^{-I125}$ peaks were also detected) in the MC-AS chromatograms (*Figure 1—figure supplement 1E*, orange panel), whereas only background radioactivity was detected in the MC-CS (*Figure 1—figure supplement 1E*, blue panel). Even in the presence of cells, no signal was detected on the MC-CS after 24 hr of adding the fluorescent dye (Alexa Fluor 594 hydrazide) in the MC-AS (*Figure 1—figure supplement 1C*). Immunofluorescent studies performed by DIV ~10, indicated that most of these neurons express the excitatory marker vesicular glutamate 1 (Vglut1) but not the inhibitory marker glutamate decarboxylase 1, Gad67 (*Figure 1—figure supplement 2*).

PCNs express MCT8 in both cellular compartments (*Figure 1A–C*). Immunostaining for MCT8 revealed that MCT8 is present in the cell bodies and short processes and the long-distant axons (*Figure 1B and C*). MCT8 colocalizes with Lamp1, a marker for endosome- and non-degradative lysosome (NDL)-like organelles (*Cheng et al., 2018*), in cell bodies and axons (inset in *Figure 1B and C*). Additionally, we looked at neurons present in the primary motor cortex of adult mice and detected MCT8 distribution to the axonal vesicles and plasma membrane through immuno-electron microscopy (*Figure 1D*). Furthermore, MCT8 was detected in the nuclear membrane and the trans and cis Golgi apparatus (*Figure 1E–G*).

The addition of $T3^{I125}$ to the MC-AS (*Figure 1H*) led to T3 uptake and transport through neuronal long distal axons. Whereas only background radioactivity was detected in the medium in the MC-CS (inset in *Figure 1I*, blue panel) after 24 hr, at later time-points (after 72 hr) about 0.5–1.0% of $T3^{I125}$ was detected in the MC-CS (*Figure 1I*, blue panel), illustrating that retrograde transport occurred, that is, MC-AS→MC-CS. Small amounts of $T2^{I125}$ and $I^{125}$ were also present, likely the result of $T3^{I125}$ metabolism. Given the concentration of T3 in the medium (2.6 nM), we estimate that between 0.8 and 1.6 fmols of T3 were retrogradely transported during the 72 hr incubation by the approximately 300 neurons that crossed the MC-CS into the MC-AS.

To test whether the retrograde $T3^{I125}$ transport was mediated via MCT8, we next added 2 µM of the highly selective MCT8 inhibitor Silychristin (SC) (*Johannes et al., 2016*) to the MC-AS and saw that it decreased the transport of $T3^{I125}$ (*Figure 1J and K*) as evidenced by the smaller size of the $T3^{I125}$ peaks detected in the MC-CS (*Figure 1J*, blue panel). We also used the same setup to test whether retrograde $T3^{I125}$ transport occurred in peripheral neurons (*Figure 1—figure supplement 3A and B*). We isolated neurons from postnatal day (P)2 rat dorsal root ganglia (DRG), which also express MCT8 (*Figure 1—figure supplement 3D*). Similar to cortical neurons, DIV ~10 DRG cells exhibited MC-AS→MC-CS transport of $T3^{I125}$, inhibited by 2 µM SC (*Figure 1—figure supplement 3C*). Altogether, these results show that T3 is retrogradely transported through axons and released on the opposite cellular compartment (MC-AS → MC-CS), with the involvement of MCT8.

Next, we wished to verify whether T3 transport in the cortical neurons also occurred in the opposite direction, i.e., MC-CS→MC-AS. Notably, adding the $T3^{I125}$ to the MC-CS (*Figure 1—figure supplement 4A*) resulted in anterograde transport of $T3^{I125}$. Whereas after 24 hr only background radioactivity was detected in the MC-AS (inset in *Figure 1—figure supplement 4B* orange panel), at later time points (72 hr) the medium in the MC-AS contained 0.4–0.8% of $T3^{I125}$ (*Figure 1—figure supplement 4B*, orange panel). The addition of SC to the MC-CS markedly reduced the amount of $T3^{I125}$ reaching the MC-AS (*Figure 1—figure supplement 4C, D*). Furthermore, adding 6 µM Xanthohumol (XH; an inhibitor of D3 [*Renko et al., 2015*]) in the MC-CS increased the MC-CS→MC-AS transport of $T3^{I125}$ (*Figure 1—figure supplement 4D*), suggesting that D3 activity in the neuronal soma is limiting the amount of T3 transported along axons.

Altogether, these results show that T3 is bi-directionally transported through axons and released on the opposite cellular compartment (MC-CS ↔ MC-AS), with the involvement of MCT8. However, despite being present in both cellular compartments, D3 only metabolized T3 in the MC-CS, limiting MC-CS→MC-AS transport of T3.

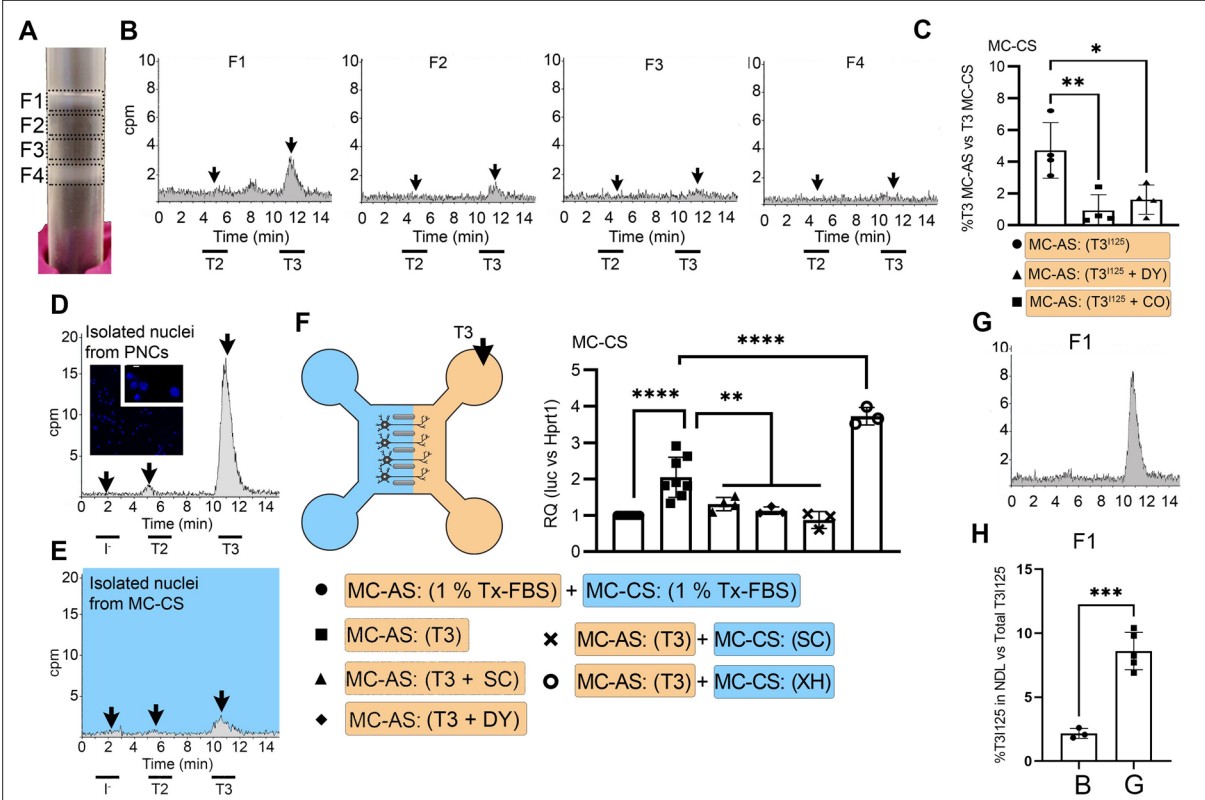

**Figure 2.** T3 is in neuronal endosomes/NDL; retrograde transport depends on clathrin-mediated endocytosis and microtubules and initiates TH signaling. (**A**) Gradient column after ultracentrifugation, the resulting four fractions are indicated. (**B**) Chromatograms of the medium after the PNCs were incubated with T3$^{I125}$ for 24 hr, and of the four fractions after ultracentrifugation. (**C**) Quantitation of T3$^{I125}$ retrogradely transported into the MC-CS medium under the indicated conditions. The Y-axis in % T3$^{I125}$ in the MC-CS vs. T3$^{I125}$ added to the MC-AS. **D**.~9 x 10$^6$ PNCs were loaded with T3$^{I125}$ for 24 hr and processed for nuclei isolation, which were subsequently studied through UPLC, showing an accumulation of T3$^{I125}$. (**E**) Same as in (**D**), except that the T3$^{I125}$ was applied in the MC-CS and the nuclei isolated from ~200.000 neurons from the MC-AS (pool of 10 microchambers). (**F**) At 8- to 10-day-old cultures were incubated for 48 hr with a medium containing 1% charcoal-stripped serum (Tx-medium; the B27 supplement is removed). Subsequently, 10 nM T3 was applied in the MC-AS for 24 hr. Bar graph shows the quantitation of the Luc mRNA levels of the cells in the MC-CS under the indicated conditions. (**G**). Same as in (**B**), except that endosomes/NDL were isolated in the presence of 2 µM SC. (**H**) Quantitation of the T3$^{I125}$ from the chromatograms found in fraction 1 in (**B** and **G**). The Y-axis in % T3$^{I125}$ in the F1 vs. T3$^{I125}$ was added to the medium. Values are mean ± SD of 3–8 independent experiments; *p<0.05, in comparison with 10 nM T3 incubation, **p<0.01, ***p<0.001 in comparison with 1 % Tx FBS incubation. SC, Silychristin; XH, Xanthohumol; Dy, Dynasore; CO, Colchicine.

## Retrograde T3 transport via neuronal endosomes/NDL

It is well known that a retrograde transport system exists in neurons based on neuronal endosomes/NDL (*Hancock, 2014*). Thus, we hypothesized that the transport of T3 uses this shuttle mechanism. To test if this was the case, we isolated and cultured DIV ~5 PCNs, which were then loaded with T3$^{I125}$ for 24 hr. Cells were then harvested and processed through iodixanol gradient ultracentrifugation for isolation of subcellular fractions containing endosomes/NDL, including fraction one (F1) that was enriched with the endosomes/NDL (*Figure 2A*). The resulting fractions were then resolved through UPLC and it was clear that most T3$^{I125}$ was contained in F1, with some spillover to F2 (*Figure 2B*).

Endosomes/NDL can be formed through clathrin-mediated endocytosis and be actively transported throughout microtubules (*Kaksonen and Roux, 2018*). To find out whether these elements were involved in the formation of endosomes/NDL containing T3, we co-incubated the MC-AS with T3$^{I125}$ and either 20 µM Colchicine (CO, an inhibitor of microtubule formation) or 80 µM dynasore (DY), an inhibitor of clathrin-mediated endocytosis (*Macia et al., 2006*). The use of CO or DY reduced the amount of T3$^{I125}$ detected in the MC-CS (*Figure 2C*). Similarly, when used with DRGs neurons, CO reduced the amount of radiation detected in the MC-CS (*Figure 1—figure supplement 3C*).

# Retrograde T3 transport initiates TH signaling in cerebral cortex neurons

It is logical to assume that some of the retrogradely transported T3 ends up in the nucleus of the neurons where it can initiate TH signaling. We first looked for T3$^{I125}$ in the nuclei of approximately 400,000 neurons after T3$^{I125}$ was added to MC-AS. These neurons include the approximately 300 neurons that crossed the bridge between MC-AS and MC-CS and mediate the retrograde transport of T3$^{I125}$. Indeed, as soon as after 24 hr, a clear peak of T3$^{I125}$ could be identified in the nuclear fraction of these neurons (*Figure 2E*). The identity of the T3$^{I125}$ peak was confirmed because it comigrated with a much more prominent T3$^{I125}$ peak obtained from nuclei of 8–9x10$^6$ cells directly labeled with T3$^{I125}$ (*Figure 2D*).

We next utilized the T3 concentration in the medium (2.7 nM) and the nuclei/medium ratio of T3$^{I125}$ (~0.0015) and estimated that the nuclei in the neurons contain approximately 0.75 ng T3/mg DNA that originated from the retrograde transport from the MC-AS. This figure is similar to what was obtained in the rat's cerebral cortex after injection of T3$^{I125}$ (*Crantz et al., 1982*), suggesting that the retrograde transport of T3 is of physiological relevance.

The presence of T3$^{I125}$ in the nuclei of the neurons indicates that the retrograde T3 transport has the potential to affect TH signaling via the initiation of T3-dependent regulation of gene expression. To test if this was the case, we next isolated cortical neurons from THAI E16.5 embryos (*Mohácsik et al., 2018*). Eight- to 10-day-old cultures were incubated for 48 hr with a medium containing 1% charcoal-stripped serum (Tx-medium; the B27 supplement was removed during this period). Subsequently, the MC-AS was incubated with 10 nM T3 (200 pM free T3), and 24 hr later, neurons in the MC-CS were harvested and processed for Luc mRNA determination (*Figure 2F*). The addition of 10 nM T3 to the MC-AS resulted in a 2.2±0.5 fold increase in Luc mRNA levels (*Figure 2F*). However, coincubation of T3 with 2 µM SC in the MC-AS significantly reduced T3 induction of Luc mRNA, highlighting the importance of MCT8 in this mechanism (*Figure 2F*).

We also tested whether the addition of DY to the MC-AS interfered with the T3 induction of Luc. Indeed, after 24 hr of the addition of T3, induction of Luc mRNA was blunted in the presence of DY (*Figure 2F*), confirming that clathrin-dependent endosomal T3 uptake is involved in T3 action in neurons. A corollary of these experiments is that T3 is taken up in the MC-AS by endosomal/

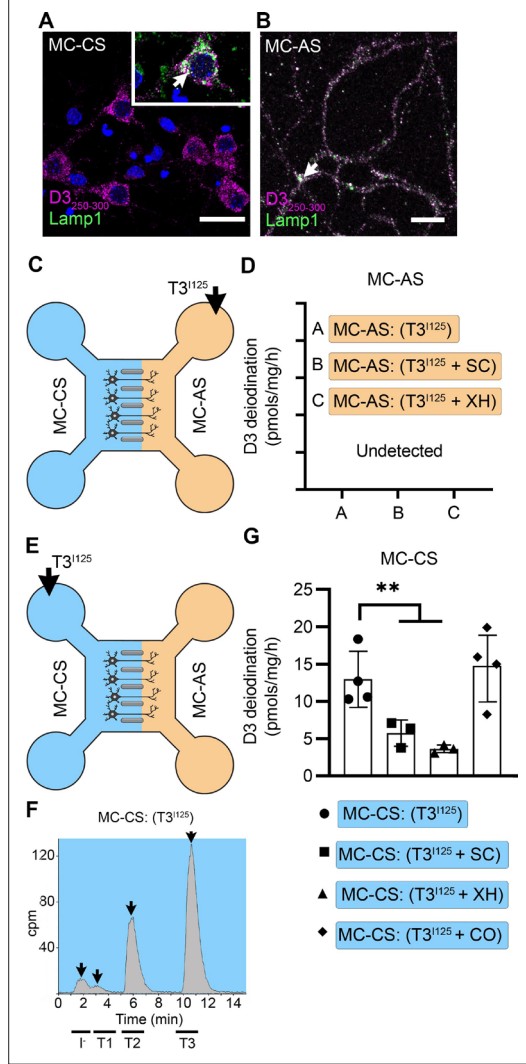

**Figure 3.** T3 is not metabolized by axonal D3. (**A**) In the MC-CS D3 staining in magenta and Lamp1 staining in green; colocalization is indicated as white and shown in the inset (arrow). Scale bar = 25 µm. (**B**). In the MC-AS, D3 staining in magenta and Lamp1 staining in green; colocalization is indicated as white (arrow). Scale bar = 25 µm. (**C,D**). No D3-mediated T2$^{I125}$ production was detected in the MC-AS. However, a high D3-mediated T2$^{I125}$ production was detected in the MC-CS (**E–G**), which was reduced by exposure to 2 µM SC or 6 µM XH but not by exposure to 80 µM DY. Values are mean ± SD of three to four independent experiments; **p<0.01 in comparison with T3$^{I125}$ incubation.

The online version of this article includes the following figure supplement(s) for figure 3:

**Figure supplement 1.** Adding SC or XH in the MC-AS did not affect the MC-CS.

NDL, transported via microtubules to the MC-CS, and released to the cell nucleus where it regulates gene expression.

## MCT8 modulates the exit of T3 from the endosomal/NDL

Not much is known about how contents in the endosomal/NDL are transferred to the cell nucleus, and here we studied whether MCT8 could have a role in this process. This was first tested in vitro by isolating T3$^{I125}$-loaded endosomes/NDL in the presence or absence of 2 µM SC. The presence of SC during isolation was associated with 4-fold retention of T3$^{I125}$ inside the F1 endosomal/NDL (*Figure 2G and H*), suggesting that MCT8 mediates the release of T3 from the endosomes/NDL. Second, we tested whether the presence of SC in MC-CS could affect TH signaling initiated by the retrograde transport of T3 from the MC-AS. While the addition of 10 nM T3 in the MC-AS doubled Luc mRNA levels after 24 hr, the presence of 2 µM SC in the MC-CS blunted Luc mRNA induction by T3 (*Figure 2F*), indicating that MCT8 plays a role in the release of T3 to the cell nucleus and initiation of TH signaling.

## T3 is not metabolized in MC-AS despite the presence of D3 in axons

Immunofluorescent studies showed that by DIV ~10, PCNs express D3 in both cellular compartments (*Figure 3A and B*). To visualize D3, we used a D3-specific antibody directed against the molecule's C-end (D3$_{250-300}$). Staining with α-D3$_{250-300}$ revealed that D3 is present in the cell bodies and short processes (*Figure 3A*) and the long-distant axons (*Figure 3B*), where the D3 signal displays a dotted pattern. This was reminiscent of our previous observations that D3 is present in early endosomes and constantly recycles with the plasma membrane (*Jo et al., 2012*; *Kalló et al., 2012*; *Baqui et al., 2003*). To test if D3 was compartmentalized in this system as well, we looked for colocalization of D3 with Lamp-1. We found that D3 co-localizes with Lamp1 in the cell bodies and axons (*Figure 3A and B*). There was also a weak D3 signal in the cell nucleus (*Figure 3A*), which is in agreement with our previous studies showing that D3 sorts to the neuronal nucleus only during hypoxic conditions (*Jo et al., 2012*).

Despite the abundant presence of D3 in the long-distant axonal network, the addition of T3$^{I125}$ to MC-AS (*Figure 3C*) for up to 72 hr revealed no deiodination of T3$^{I125}$; only background amounts of T2$^{I125}$ were detected in the medium, equivalent to when no cells were added. Moreover, the addition of 6 µM of the D3 inhibitor XH to the MC-AS also did not affect the retrograde transport of T3$^{I125}$ (*Figure 1K*). D3 is a transmembrane protein with the active center of the enzyme located in the cytosol (*Kalló et al., 2001*). Thus, it is conceivable that most T3 that is taken up by axons ends up in the endosomal/NDL vesicles rather than in the cytosol, where it would be easily metabolized by D3.

A different scenario altogether was identified in the body of the neurons. The addition of T3$^{I125}$ to the MC-CS (*Figure 3E*) resulted in a prominent peak of T2$^{I125}$, which reflects the uptake of T3$^{I125}$ to the cytosol, metabolism, and release of T2$^{I125}$ to the medium (*Figure 3—figure supplement 1A and B*). The rate of T3 metabolism was high, consuming approximately ¼ of the added T3$^{I125}$ at every 24 hr. The addition of 2 µM SC or 6 µM XH to the MC-CS markedly reduced the metabolism of T3$^{I125}$ (*Figure 3G*), evidenced by the decrease in the peaks of T2$^{I125}$ found in the MC-CS. As expected, the addition of SC or XH in the MC-AS did not affect the metabolism of T3$^{I125}$ in the MC-CS (*Figure 3—figure supplement 1A, B*), indicating that these drugs are not transported across compartments. These data suggest that T3 entering the neuronal cell body via MCT8 is rapidly targeted for inactivation via D3.

## D3 in the MC-CS modulates TH signaling initiated by retrograde transport of T3

The fact that T3 in the endosomal/NDL vesicles is transferred to the cell nucleus to initiate TH signaling in the neighborhood of high D3 activity in the neuronal cell bodies raises the possibility that those D3 enzymes could modulate TH signaling. This was investigated by measuring TH signaling initiated by retrogradely transported T3 in the presence or absence of 6 µM XH in the MC-CS. Remarkably, XH enhanced the T3 induction of Luc mRNA levels by 1.8-fold (*Figure 2F*). These results suggest that D3 activity in the neuronal soma, but not in the axons, limits the amount of T3 that is transferred from the endosomal/NDL vesicles to the cell nucleus.

## T3 transport triggers localized TH signaling in the mouse brain

In the next set of experiments, we looked for in vivo evidence of axonal transport of T3 in brain areas in which neurons express both MCT8 and D3. First, we looked at T3$^{I125}$ transport in the medial basal hypothalamus while revisiting our previous hypothesis that there is a retrograde axonal transport of T3 from the median eminence to the hypothalamic paraventricular nucleus (PVN). This could explain how T3 generated by tanycytic D2 can down-regulate TRH expression (*Kalló et al., 2012*) or how T3 content in the ME can reduce PVN TRH expression independently of circulating TH levels (*Sinkó et al., 2023*). Here, we experimentally tested this by injecting T3$^{I125}$ directly into the hypothalamic median eminence (ME) of rats. Thirty minutes later, radioactivity was detected in the PVN, whereas only background activity was found in the lateral hypothalamus and cerebral cortex (*Figure 4A–C*).

Second, we looked at axonal T3 transport and TH signaling using the TH-action indicator THAI transgenic mouse (*Mohácsik et al., 2018*). Primary motor cortex neurons (M1 neurons) of this mouse model express D3 and MCT8 (*Figure 4D–H* and *Figure 4—figure supplement 1*), a cortical area that is highly responsive to T3 (*Figure 4—figure supplement 2A and B*). For the experiment, T3 crystals were stereotaxically implanted into the right hemisphere at the level of the M1 (*Figure 4I*). After 48 hr of implanting the T3 crystals, we not only found local induction of Luc mRNA (2.7±1.2 fold) at the site of implantation but also induction in the contralateral M1, which receives interhemispheric axonal projections through the corpus callosum (*Figure 4J*). We also detected T3 signaling in the ipsilateral secondary somatosensorial cortex (S2) that likewise receives projections from M1 (*Paxinos, 2004*). The absence of induction of two T3-responsive markers in the hypothalamus, Luc (*Figure 4K*), and the TRH-degradation enzyme (trh-de; *Figure 4—figure supplement 2C*), a highly T3-sensitive region not connected directly with M1, indicates that the implanted T3 molecules did not diffuse randomly. These findings support the hypothesis that T3 can be selectively transported along neuronal axons and can, therefore, initiate TH signaling in distant but discrete brain areas.

## Discussion

The discovery that the Allan-Herndon-Dudley syndrome is caused by mutations in MCT8 revealed that transport mechanisms across cell membranes were involved in TH action in the brain (*Dumitrescu et al., 2004*; *Friesema et al., 2004*). However, critical questions remained unresolved. Here we address some of those questions and provide the mechanistic basis for the T3 signaling in the brain. We now show that neurons utilize the coordinated expression of MCT8 and D3 to create an unimpeded pathway for T3 to reach the cell nucleus that starts at the axonal termini, where T3 molecules are concentrated into endosomes/NDL. These T3-loaded vesicles are retrogradely transported to the neuronal cell body, delivering T3 to the nucleus where it regulates gene expression. T3 molecules that escape the endosomes/NDL prematurely or enter the cytosol directly from the extracellular space are rapidly inactivated via D3.

Three steps characterize the retrograde axonal transport of T3: First, T3 is loaded into clathrin-dependent MCT8- and D3-containing endosomes/NDL in long distal axons; once inside the endosomes/NDL, T3 is protected from D3-mediated catabolism because D3's active center is cytosolic (*Kalló et al., 2012*). This is illustrated by the fact that there is no T3 metabolism in MC-AS, a condition that is not affected by adding XH to the MC-AS (*Figure 3C and D*). Second, the T3-containing endosomes/NDL travel retrogradely through a microtubule-dependent mechanism, which is illustrated by the fact that adding CO to the MC-AS decreases by >50% the amount of T3 found in the MC-CS (*Figure 2C*; *Figure 1—figure supplement 3C*). Third, T3 exits the endosomes/NDL and reaches the nuclear compartment to establish a transcriptional footprint. Indeed, we were able to identify retrograde transport of T3$^{I125}$ to the neuronal nuclei (*Figure 2E*). T3 molecules that bypass this pathway and enter the cytosol directly through MCT8 are subject to active D3-mediated inactivation to T2 (*Figure 3E–G*).

In neurons, signaling endosomes/NDL are normally organized at the axonal termini, adjacent to the post-synaptic membrane or glial cells. These endosomes/NDL are retrogradely transported via microtubule-dependent dynein motors from the distal end of a long axon to the cell body, enabling extracellular molecules to modify gene expression (*Cosker and Segal, 2014*). Endosomes/NDL direct molecular cargo along four main routes: recycling to the cell surface, transport to the Golgi apparatus, degradation in endolysosomes, or transport to the nucleoplasm. The presence of small amounts of

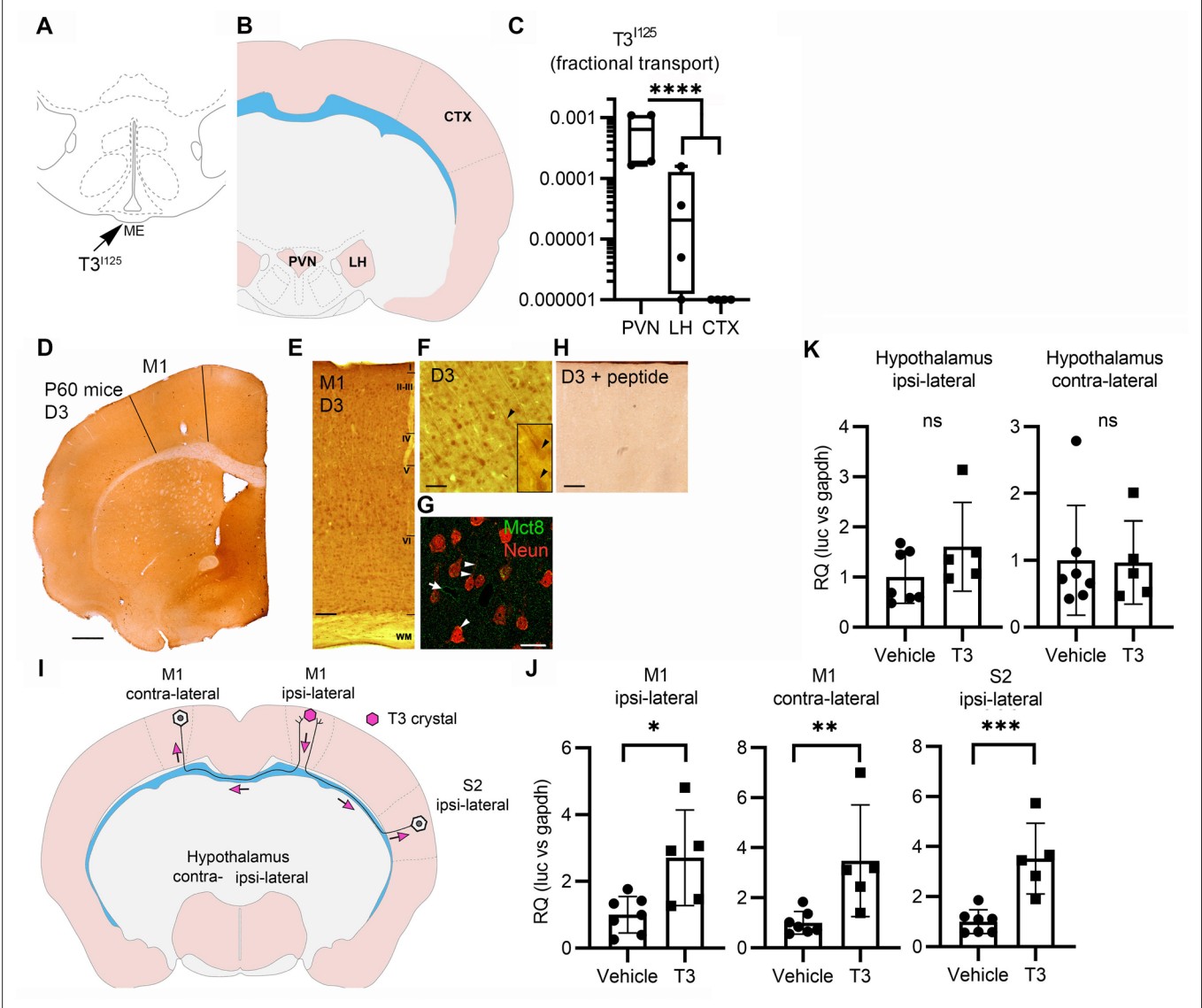

**Figure 4.** Axonal T3 transport can initiate TH signaling between two interconnected brain areas. (**A**) Cartoon showing the injection site of T3$^{I125}$ in the median eminence (Bregma – 2.56 mm). Scale bar = 1 mm. (**B**) The areas dissected after 30 min of injection are indicated (Bregma –1.80). (**C**) Quantitation of the transported T3$^{I125}$ from the rat median eminence to the indicated areas (**D**) D3 immunostaining in the adult mouse M1 cortex. (**E**) Dio3 immunoreactive neurons were found in all layers of the M1 cortex. Scale bar = 150 µm. (**F**) Detail of layer V, where pyramidal neurons and their apical dendrites were stained (arrowheads). Scale bar = 25 µm. The bar in 4F represents 25 µm in the inset in 4F. (**G**) Immunofluorescence of M1 neurons in the P60 mice cortex using the indicated antibodies. Mct8 is evenly distributed in the M1, including in apical dendrites of neurons (arrowhead), and showed higher intensity in the wall of the capillaries (arrow). Scale bar = 10 µm. (**H**) Incubating the α-D3$_{250-300}$ with its blocking peptide resulted in no staining. Scale bar = 25 µm. The bar in 4F represents 25 µm in the inset in 4F. (**I**) A T3 crystal (pink hexagon) was inserted into the M1 of the right hemisphere, which receives axons from neurons located in the indicated areas. The pink arrows indicated the direction of the transported T3. (**J**) Quantitation of Luc mRNA levels at the indicated brain areas. (**K**) Hypothalamic expression of Luc mRNA. Values are mean ± SD of four to seven independent experiments; *p<0.05, **p<0.01, ***p<0.001; ns: non-significant.

The online version of this article includes the following figure supplement(s) for figure 4:

**Figure supplement 1.** Staining with α-D3$_{250-300}$ resulted in a similar spatial expression of D3 (in the P11 rat cortex) as previously reported for its mRNA with in situ hybridization (*Escámez et al., 1999*).

**Figure supplement 2.** T3 triggers TH signaling in the brain of the THAI mice (**A**).

MCT8 (*Figure 1E and F*) and D3 (*Freitas et al., 2010*; *Jo et al., 2012*) in the nuclear membrane suggests that the latter mechanism is involved in the transfer of T3 to the cell nucleus, allowing for extracellular signals to affect nuclear events such as gene expression (*Chaumet et al., 2015*). Our data indicate that not only T3 is cargo to these endosomes/NDL but also that T3 uses this transport system to regulate gene expression (*Figure 2F*).

The widespread MCT8 expression in the brain (*Bernal et al., 2015*; *Vatine et al., 2017*; *Wang et al., 2023*) supports a role for MCT8 in neuronal T3 signaling, but the exact mechanism remained elusive. Studies using iPSC-derived neural cells suggest that MCT8 might have a greater role at the BBB, a view that is also supported by studies in zebrafish and mice (*Vatine et al., 2017*; *Ceballos et al., 2009*; *Zada et al., 2016*; *Mayerl et al., 2014*). However, these studies did not consider T3 metabolites, which are certain to be generated given the high D3 activity in neural cells (*Figure 3F*). In addition, there are discrepancies between human and animal models because of the expression of alternative TH transporters in the latter (*Vancamp and Darras, 2018*). Thus, a unifying hypothesis for the contribution of MCT8 to TH signaling in neural cells and how its loss-of-function mutations relate to the neurological manifestations seen in patients was missing.

The present studies identified two important roles played by MCT8 in murine cortical neurons. First, MCT8 is critical in the endosomes/NDL pathway that retrogradely transports T3, including the exit of T3 from the vesicles (*Figure 2F–H*), which can then enter the cell nucleus and affect gene transcription. Second, as in other cells, MCT8 mediates T3 transport into the cytosol (cell bodies; *Figure 3G*).

The brain of a healthy, non-pregnant adult exhibits the highest D3 activity level (*Hernandez et al., 2006*). Nonetheless,>90% of the TRs in the brain are occupied with T3 (*Crantz et al., 1982*), a figure much higher than any other tissue — TR occupancy with T3 in the liver is about 50%. Thus, the present findings resolved this paradox, explaining how a pathway that takes advantage of the topological orientation of D3 (catalytic active center facing the cytosol) avoids the catabolism of T3 that is incoming through the endosomes/NDL. The physiologic implications of these findings are considerable, as they reveal potential checkpoints for TH signaling in neurons. For instance, we had previously observed that in hypoxic neurons D3 accumulates in the nuclear membrane, reducing TH signaling (*Jo et al., 2012*). We now show that inhibiting D3 in the cell body (nucleus) enhances the transfer of T3 from the endosomes/NDL to the cell nucleus and stimulates gene expression (*Figure 2F*). Thus, under certain pathological conditions (e.g. hypoxia), the entry of T3 in the nucleus can be regulated by the presence of D3.

The present studies also advance our understanding of how T3 in the median eminence (plasma-born and locally D2-generated) may regulate TRH expression in the PVN, expanding on our original hypothesis (*Kalló et al., 2012*). The speed with which injected T3 traveled retrogradely to the PVN—just 30 min—is remarkable. In addition, the present studies brought to light the unanticipated reality that T3 molecules can be taken up by long neurons and transported to distant locations in the brain. In fact, we found that T3 originating from one brain hemisphere was taken up, transported, and had an effect on gene expression in neurons located in the contralateral hemisphere (*Figure 4*,I,J). Of note, the intensity of the T3-induced Luc expression varied between ~2.7- and 3.4-fold (M1 ipsilateral and contralateral; *Figure 4J*), which is within the fold-range observed during the transition between hypothyroidism and TR saturation in the cerebral cortex (2.5-fold; *Figure 4—figure supplement 2B*). This suggests that the axonal pathway that retrogradely transports T3 operates well within the physiological context.

The present study can be expanded in the future to address some important remaining gaps. The mechanism through which T3 is taken up and concentrated within the endosomes/NDL has not been established. Likewise, it is unclear whether the anterograde transport of T3 serves a physiological purpose, or it is a byproduct of endosomes/NDL recycling. Of note, previous studies have suggested that T3 can function as a noradrenergic neurotransmitter and be delivered to the synaptic cleft (*Dratman and Gordon, 1996*; *Gordon et al., 1999*). Despite clear evidence of MCT8 in the nuclear membrane (*Figure 2E*), details of the transfer of T3 from the endosomes/NDL to the nucleus need to be clarified. Lastly, the possibility that other TH transporters, for example LAT1-2, play similar roles as MCT8 needs to be investigated. This is less likely to be relevant given that SC fully prevented the T3 (MC-AS→MC-CS) induction of Luc (*Figure 2F*).

Our study presents several limitations: (i) the in vitro model contained excitatory cortical (central) and dorsal root ganglia (peripheral) neurons, and the in vivo model contained cortical and hypothalamic neurons, suggesting that axonal transport of T3 is not confined to specific neuronal subtypes. However, considering the high number of neuronal subtypes, we cannot rule out that different mechanisms of T3 transport exist. This could be clarified in future studies using genetically modified animal models; (ii) we have not studied T3 transport in selected neurosecretory PVN neurons projecting to the ME, thus we have not unequivocally proven that the T3 retrograde transport in the ME is present in TRH neurons. But vasopressin and oxytocin neurons projecting axons do not terminate in the ME. That the microinjected T3 in the ME could be transported by other neuroendocrine neurons located in the periventricular nucleus —such as CRH, vasopressin, oxytocin, and somatostatin neurons—in addition to hypophysiotropic TRH neurons, must also be considered.

The present findings reveal that T3 molecules entering neurons directly from the extracellular compartment can be rapidly inactivated to T2. In contrast, those T3 molecules that are selectively taken up into clathrin-dependent, MCT8- and D3-containing endosomal/NDL, are protected against degradation during the transport to the cell nucleus. There, perikaryon D3 modulates the entry of T3 molecules transitioning from the endosomes/NDL and those entering the cytosol directly from the extracellular space. Altogether, the present findings resolve the paradox of the high T3 nuclear content in the brain amid a very high level of D3 activity. They also explain how therapy for hypothyroidism that contains L-T3 can bypass the neuronal D3 catabolism and safely reach the neuronal nucleus to restore TH signaling.

## Methods

All experiments were approved by the Institutional Animal Care and Use Committee at the University of Chicago (#72577) or by the Animal Welfare Committee at the Institute of Experimental Medicine and followed the American Thyroid Association Guide to investigating TH economy and action in rodents and cell models (*Bianco et al., 2014*).

### T3$^{I125}$ injection in the median eminence

220 g adult Wistar male rats were kept on a heating pad while undergoing ventral transsphenoidal surgery (*Reg, 1991*) to expose the median eminence (ME) of the hypothalamus. The T3$^{I125}$ was freshly purified by LH-20 column and applied to the ME using a Nanoliter 2010 microinjector. ~40,000 cpm were injected in a volume of 50 nl. After 30 min, the animals were killed, the brain was removed and PVN, the lateral hypothalamus (LH), and a cortical sample were microdissected with the Palkovits's punch technique (*Egri et al., 2016*) and counted in a gamma counter.

### T3 signaling in the THAI mouse brain

2-month-old male THAI mice were injected with 0.1 and 1 μg/BW T3 i.p. and decapitated 24 hr later. The cerebral cortex was dissected and assayed for Luc activity as described (*Mohácsik et al., 2018*) using an assay system reagent (Promega, Madison, WI) on a Luminoskan Ascent luminometer (Thermo Electron Corp. Labsystems, Vantaa, Finland); the relative light unit (RLU) was normalized to protein content. To study T3 trafficking across different cortical areas, we inserted crystalized T3 (*Lechan and Kakucska, 1992*) into the M1 of one hemisphere using stereotaxic surgery; control mice were sham-operated. Animals were decapitated after 48 h. The M1 and the contralateral M1, the ipsilateral S2, and the ipsi- and the contralateral portion of the hypothalami were microdissected and processed for RT-qPCR using TaqMan Real-Time.

### Primary embryonic cortical neurons were cultured in a microfluidic compartmentalized device

PCNs were isolated from E16.5 THAI mouse embryos. Briefly, embryos were removed, and the cerebral cortex dissected, stripped of meninges, and dissociated into a combination of Ca$^{2+}$ and Mg$^{2+}$ free Hanks balanced salt solution (HBSS) containing 0.25% trypsin-0.53 mM EDTA, then mechanically triturated using fire-polished glass Pasteur pipettes. Isolated cells were passed through a 40 μM cell strainer to reach a cell density of 4.5×10$^6$ cells/ml. We used a microfluidic compartmentalized culture device that contains 450 μM long microchannels connecting MC-CS and MC-AS and permits only

distal axons to grow into the MC-AS (XonaChip, Cat# XC450, Xona Microfluidics Temecula, CA, USA). The cortical neurons were plated at a density of 5–9 x $10^4$ cells/device in the cell compartment with a growth medium composed of the neurobasal medium, 2% B-27 supplement, 1% GlutaMax, and 1% antibiotic-antimycotic (Penicillin-streptomycin; all from Gibco). On DIV ~2, one-half of the medium was replaced with a growth medium containing the anti-mitotic cytosine arabinoside (Sigma-Aldrich) which restricts astrocytes and microglia to <0.01% (*Hasel et al., 2018*). Thereafter, the growth medium was replaced every other day. During the different experiments, each compartment was fluidically isolated by hydrostatic pressure, accomplished by keeping the medium volume in one compartment higher than in the opposite compartment, allowing us to differentially treat either side (*Park et al., 2006*). For the isolation of DRG neurons, the dorsal root ganglia were dissected and cultured from 2-day-old Wistar rats and THAI 6-day-old THAI mice according to published protocols (*Campenot et al., 2009*; *Watson et al., 1999*). All other procedures were as with the PCNs.

## Cell staining and Immunofluorescence studies

Cultures at DIV ~10 were fixed in 4% paraformaldehyde for 20 min, washed twice in PBS, and then permeabilized in PBS with 0.1% Triton X-100 for 5 min. Cultures were blocked in PBS with 5% BSA for 15–30 min at room temperature and incubated with primary antibody diluted in PBS, at 4 °C overnight. Cultures were rinsed 3 times and incubated for 2 hr at room temperature with a secondary antibody. Primary and secondary antibody dilutions were as follows: mouse monoclonal anti-Lamp1 antibody 1:1000 (Biotechne; AF4320), rabbit polyclonal anti-MCT8 antibody 1:400 (Atlas antibodies; HPA003353), rabbit polyclonal anti-D3 antibody (Novus Biologicals; NBP1-06767), rabbit polyclonal anti-GFAP (1:250), guinea pig polyclonal anti-NeuN (Millipore; ABN90), rabbit polyclonal anti-Gad67 (Invitrogen; PIPA585371), rabbit polyclonal anti-Vglut1 (Invitrogen; PIPA585764), alexa 488 conjugated goat anti-mouse IgG 1:200 (Vector), alexa 594 conjugated horse anti-rabbit IgG 1:200 (Vector), alexa 488 conjugated horse anti-rabbit IgG 1:200 (Vector), Cy3 anti-guinea pig. The images were analyzed by NIS-Element AR (Nikon Instruments) or ImageJ software (NIH). Final Figures were prepared on Adobe Photoshop.

## Immuno-electron microscopy for Mct8

Mice were anesthetized with a mixture of ketamine and xylazine (50 and 10 mg/kg BW i.p., respectively), perfused (trans-cardiac) with 10 ml 0.01 M phosphate-buffered saline (PBS), and fixed with 40 ml of 2% paraformaldehyde and 4% acrolein in 0.1 M phosphate buffer (PB). The brains were removed and postfixed by immersion in 4% PFA in PBS overnight at room temperature. Coronal 25-μm-thick sections containing the primary somatosensorial cortex were cut with a vibratome (Leica VT 1000 S) and stored at –20 °C in 30% ethylene glycol, 25% glycerol in 0.05 M PB until further use. Pretreatment included 30 min incubation with 1% sodium borohydride and 15 min with 0.5% $H_2O_2$, followed by a sucrose gradient (15 → 30 %) and three frozen-thaw cycles in liquid nitrogen. For immunohistochemistry, sections were blocked with 2% normal horse serum and incubated with rabbit polyclonal antiserum against MCT8 (1:20,000; kind gift of Dr. TJ Visser) for 4 days at 4 °C, followed by biotinylated donkey anti-rabbit IgG (1:200; Jackson Immuno Research Labs,) for two hours and 0.05% DAB / 0.15% Ni-ammonium-sulfate / 0.005% $H_2O_2$ in 0.05 M Tris buffer (pH 7.6). The staining was silver-gold-intensified using the Gallyas method (*Kalló et al., 2001*; *Liposits et al., 1984*). For electron microscopy, sections were incubated in 1% osmium-tetroxide for 1 hr at room temperature and then treated with 2% uranyl acetate in 70% ethanol for 30 min. Following dehydration (ethanol - acetonitrile) the sections were embedded in Durcupan ACM epoxy resin on liquid release agent coated slides and polymerized at 56 °C for 2 days. Ultrathin, 60–70 nm-thick sections were cut with Leica UCT ultramicrotome (Leica Microsystems, Vienna, Austria), were mounted on Formvar coated, single-slot grids, and treated with lead citrate. Images were obtained using a transmission electron microscope (JEOL-100 C). For the experiments with rats (*Figure 4—figure supplement 1*), immunostaining was performed according to previously published studies (*Navarro et al., 2019*). Coronal 100 μm thick sections were cut with a vibratome (Microm HM650V; Thermo Fisher) and stored in PBS-azide until further use. Sections were incubated overnight with rabbit anti-deiodinase type 3 (Dio3) polyclonal antibody (1:400, Novus), followed by biotinylated goat anti-rabbit antibody (1:200), Vectastain ABC kit (1:200, Vector Laboratories), and 0.05% 3,3´ diaminobenzidine (DAB, Sigma-Aldrich).

## Lysosomes isolation by ultracentrifugation

We used a lysosome enrichment kit (Thermo Scientific) and followed the manufacturer's instructions. Briefly, approximately 50–200 mg of cells were harvested and lysed using a sonicator (15 bursts; 9 W power). Subsequently, the homogenate was centrifuged to remove cellular debris. The resulting supernatant was overlayed on several discontinuous gradients of the OptiPrep Cell Separation Media (60% iodixanol in water with a density of 1.32 g/ml) and ultracentrifuged (145,000 g for 180 min at 4 °C) to isolate and enrich for lysosomes. The different fractions were removed from the gradient, pellet by centrifugation, and washed three times with PBS. The final pellet was lysed in 30 µl of 0.02 M ammonium acetate +4% methanol +4% PE buffer and studied through UPLC.

## Iodothyronine chromatography using UPLC

PCNs at DIV ~10 were incubated with $10^6$ cpm of $T3^{I125}$/ml, totaling 60 µl per well (total two wells). After 24 and 72 hr, 100 µl of the medium was sampled, mixed with 100 µl of 0.02 M ammonium acetate +4% methanol +4% PE buffer (0.1 M PBS, 1 mM EDTA), and applied to the UPLC column (AcQuity UPLC System, Waters). Fractions were automatically processed through a Flow Scintillation Analyzer Radiomatic 610TR (PerkinElmer) for radiometry. The D3-mediated deiodination was calculated by the production of $T2^{I125}$ /h / mg protein (*Boucai et al., 2022*).

## TaqMan real-time quantitative PCR

Total RNA was isolated from microdissected specific brain areas with NucleoSpin RNA kit (Macherey-Nagel)) or from neurons growing in MC-CS with an RNeasy Mini kit (Thermo Fisher). DNA contaminants were digested with DNASE I (Ambion). Undiluted total RNA (1 µg) was reverse transcribed with the High-Capacity cDNA Reverse Transcription Kit (ThermoFisher Scientific, Waltham, MA. cDNA concentration was determined with Qubit ssDNA assay kit and 10 ng cDNA was used in each Taqman reaction. Luciferase expression was detected using a specific TaqMan probe (Applied Biosystems; Assay ID: AIY9ZTZ) using TaqMan Fast Universal PCR Mastermix (Applied Biosystems) and compared with hypoxanthine phosphoribosyltransferase 1 (Hprt1; Mm01545399) or Glyceraldehyde 3-phosphate Dehydrogenase (Gapdh; Mm99999915) housekeeping genes expression. Reactions were assayed on a Real-Time PCR instrument (Applied Biosystems, Waltham, MA). For *trh-de* gene, we used the #Mm00455443_m1 Taqman probe.

## Statistics

All data were analyzed using Prism software (GraphPad). Unless otherwise indicated, data are presented as scatter plots depicting the mean ± SD. Comparisons were performed by a two-tailed Student's t-test, and multiple comparisons were by ANOVA followed by Tukey's test. A $p < 0.05$ was used to reject the null hypothesis.

## Acknowledgements

The authors are grateful to support from NIDDK DK58538, DK65055, the Hungarian National Brain Research Program 2; FS-L was supported in part by grant DK15070. The technical help of Andrea Juhász and Dóra Fazekas is gratefully acknowledged. ACB is a consultant for AbbVie, Synthonics, Sention, and Thyron.

## Additional information

### Competing interests

Antonio C Bianco: Consultant fees: AbbVie, Synthonics, Sention, Thyron, Accella. The other authors declare that no competing interests exist.

## Funding

| Funder | Grant reference number | Author |
|---|---|---|
| National Research, Development and Innovation Office | The Hungarian National Brain Research Program 2 | Csaba Fekete Balázs Gereben |
| National Institute of Diabetes and Digestive and Kidney Diseases | DK58538 | Balázs Gereben |
| National Institute of Diabetes and Digestive and Kidney Diseases | DK58538 | Antonio C Bianco |
| National Institute of Diabetes and Digestive and Kidney Diseases | DK65055 | Antonio C Bianco |
| National Institute of Diabetes and Digestive and Kidney Diseases | DK15070 | Federico Salas-Lucia |

The funders had no role in study design, data collection and interpretation, or the decision to submit the work for publication.

## Author contributions

Federico Salas-Lucia, Conceptualization, Formal analysis, Investigation, Visualization, Methodology, Writing – original draft, Writing – review and editing; Csaba Fekete, Conceptualization, Formal analysis, Supervision, Funding acquisition, Investigation, Methodology, Writing – review and editing; Richárd Sinkó, Péter Egri, Kristóf Rada, Yvette Ruska, Formal analysis, Investigation, Methodology; Balázs Gereben, Conceptualization, Resources, Data curation, Formal analysis, Supervision, Funding acquisition, Validation, Investigation, Methodology, Writing – original draft, Project administration, Writing – review and editing; Antonio C Bianco, Conceptualization, Resources, Data curation, Software, Formal analysis, Supervision, Funding acquisition, Validation, Methodology, Writing – original draft, Project administration, Writing – review and editing

## Author ORCIDs

Federico Salas-Lucia 
Csaba Fekete 
Balázs Gereben 
Antonio C Bianco 

## Ethics

All experiments were approved by the Institutional Animal Care and Use Committee at the University of Chicago (#72577) or by the Animal Welfare Committee at the Institute of ExperimentalMedicine and followed the American Thyroid Association Guide to investigating TH economy and action in rodents and cell models (52).

## Decision letter and Author response

Decision letter https://doi.org/10.7554/eLife.82683.sa1
Author response https://doi.org/10.7554/eLife.82683.sa2

# Additional files

## Supplementary files
• MDAR checklist

## Data availability

All data generated or analyzed during this study are included in the manuscript and supporting file; Source Data files have been provided for Figure 1—figure supplement 3.

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
