## [Editor Report]

This novel study by Salas- Lucia examines retrograde transport of T3 in neurons using a compartmentalized microfluid device in-vitro and implantation of T3 crystals in the vivo models to understand the cellular mechanisms of T3 transport and activity in neurons. Furthermore, the authors show how T3 transport by this non-degradative lysosomal mechanism would activate genes in the nucleus. The experiments are well-designed and support the results and conclusions.

---

## [Decision Letter]

**Decision letter after peer review:**

Thank you for submitting your article "A pathway for T3 signaling in the brain to improve the variable effectiveness of therapy with L-T4" for consideration by *eLife*. Your article has been reviewed by 3 peer reviewers, and the evaluation has been overseen by a Reviewing Editor and Mone Zaidi as the Senior Editor. The reviewers have opted to remain anonymous.

Essential revisions:

This is a potentially interesting study on understanding T3 signaling in the brain and the therapeutic implications of studying the signaling pathways. Although the experiments and methodological approach used for the study are extensive, however, the enthusiasm of all three reviewers of this manuscript was low due to several weaknesses in some of the experiments and also the disjoint between the first part and second part of the study. As a result of this, the manuscript cannot be accepted in its current form. We look forward to receiving a fully revised version of the manuscript with additional experiments, and revisions as suggested by the reviewers.

*Reviewer #1 (Recommendations for the authors):*

In conclusion, there are a lot of weaknesses in the conclusions of a very ambitious study. I believe that it would be better to split it into two articles. My feeling is that the connection between the two parts (deiodination and transport) is artificial, while the experiments presented in Figures 3 and 4 represent the most promising part of the article. It would be important to reinforce each part by using KO mice, or an alternative genetic strategy, to reinforce the conclusions.

*Reviewer #2 (Recommendations for the authors):*

The main concerns raised by this reviewer were detailed in the public review. Here I make additional suggestions for potential improvements.

1) Furthermore, experiments in Figure 4 seem to present an average of only two independent experiments, which are not sufficient to draw robust conclusions.

2) It would be helpful to include a paragraph discussing the limitations of this study in the Discussion.

3) Figure 2: The hippocampus is enriched in GFAP-positive astrocytes, whereas some other brain regions (e.g. the cortex) are not so (e.g. 10.1155/2019/9605265). Although not the main topic of study here, these different astrocyte phenotypes may result in different outcomes when comparing the cortex and hippocampus. Further, do hippocampal astrocytes present altered reactivity in Thr92Ala-DIO2 mice? A few sentences in the discussion could be interesting.

*Reviewer #3 (Recommendations for the authors):*

As stated in the public review, although 2 topics that they deal with in this manuscript are interesting, it may not be appropriate to deal with two completely different topics in one paper. I rather suggest deleting table one, Figure 1, and Figure 2, and re-write the paper with other data. Even after such a modification, this paper is still very attractive, although several additional experiments may be required.

It is interesting to examine retrograde axonal transport using a compartmentalized chamber. However, since primary cortical neurons contain a different subset of neurons, particularly excitatory (glutamatergic) and inhibitory (GABAergic) neurons, it is necessary to characterize further whether this transport is specific to a certain subset of neurons or ubiquitous to all of them.

In the same line, regarding the microinjection of T3 into the median eminence, several different neuroendocrine neurons are located in the periventricular nucleus such as CRH, vasopressin, oxytocin, and somatostatin neurons, in addition to TRH neurons. Thus, it is necessary to characterize further whether retrograde transport is seen in TRH neurons to prove their hypothesis.

Although I suggest deleting table 1, I have a major comment regarding this table. In this table, the changes in LUC mRNA were indicated by arrowheads. Thus, it is rather difficult to compare the pattern of changes between Thr92 and Ala92 mice. Relative levels of LUC mRNA should be shown.

[Editors' note: further revisions were suggested prior to acceptance, as described below.]

Thank you for resubmitting your work entitled "Axonal T3 Uptake and Transport Triggers Thyroid Hormone Signaling in the Brain" for further consideration by *eLife*. Your revised article has been evaluated by Mone Zaidi (Senior Editor) and a Reviewing Editor.

The manuscript has been improved and previous comments have been well addressed, however, there are some remaining issues that need to be addressed, as outlined below by the reviewers. Although the general agreement is that the study is novel in addressing retrograde transport of T3 in neurons and the experiments are well designed to address them but there are some aspects that reviewers have pointed out which should be addressed in the revised submission.

*Reviewer #2 (Recommendations for the authors):*

This revised manuscript is much more straightforward and concise. The authors have adequately addressed most of the comments I had. This reviewer has no more suggestions for this manuscript.

*Reviewer #4 (Recommendations for the authors):*

Salas-Lucia et al. investigated novel Triiodothyronine (T3) transport and action in neurons with an in vitro system (compartmentalized microfluid device) and an in vivo experiment (implantation of T3 crystals into the brain cortex). The current version of the manuscript was updated according to new experiments based on previous comments. Although it still shows exciting data for this field, some concerns still exist as below.

1) In the previous version of the manuscripts, the authors found retrograde T3 transport in neurons (Figure 2I) as a novel finding. On the other hand, the new experiment (Figure 2-S3B) also showed the same rate of anterograde T3 transport from MC-CS to MC-AS. Also considering that MCT8 is expressed in every part of neurons, T3 is transported into neurons everywhere and by both anterograde and retrograde way in the same rate. So, T3 in neurons is homogenously distributed and circulated by bidirectional transport through microtubules. According to those results, I agree that the retrograde transport T3 in neurons is a novel finding, but I feel that the impact of this finding is not striking.

2) In the first part of the experiment, [1]T3125 is added to MA-AS, [2] T3125 is transported into neurons through MCT8 and transported to cytosol. [3] some T3125 is converted T2 by Dio3 and some other T3125 is transported again (by MCT8) and appeared in MC-CS. Although the authors could detect the T3125 in such as Figure 2 IJ(orange panel) and Figure 2-SBC(blue panel), but its levels are very low. This data raises doubts about accuracy. So I would suggest that how the concentration of T3125 in neuron without medium at MC-CS. MCT8 locating in MC-CS may significantly limit the transport of T3125 from neuron to medium.

3) In Figure2-S2C, the authors confirmed the results in Figure2 by using rat DRG neurons. The order of Y-axis is completely different from Figure 2. How do the authors interpret it?

(Also, Figure 2-S2 should be described in a similar fashion to Figure 2. Please describe the meaning of the asterisk, too.)

4) In the second part, the authors implanted T3 crystals or T3 labeled by radioactive iodine to prove the retrograde transport of T3 in rat brain. As reviewer 1 pointed out in the previous comments, it is difficult to interpret. Because there is no direct evidence of retrograde T3 transport in Figure 5. Although the authors showed no T3 effects on both ipsi- and contr- lateral hypothalamus (Figure 5K), these are still conclusions by exclusion.

*Reviewer #5 (Recommendations for the authors):*

This work used in vitro and in vivo approaches to understand thyroid hormone signaling at cellular resolution. The results show that T3 is transported by the thyroid hormone transporter MCT8 into neuronal axons. In the neurons, it is transported to the cell nucleus and activates gene expression. Type 3 deiodinase (D3) inactivates T3 signaling in specific cellular localization. These findings provide a cellular mechanism for T3 transport and activity, which is supported by experiments in both compartmentalized microfluid device and mouse systems.

The goal of this work is to study thyroid hormone (T3) transport and signaling at cellular resolution. Salas‐Lucia and colleagues used two models – a microfluid device and mice. They showed that T3 enters into neuronal axons and transport by non-degradative lysosomes (NDLs) to the nucleus, where it activates thyroid responding genes. This process is regulated by the thyroid transporter MCT8 and type 3 deiodinase (D3). In addition, the findings explain how T3 escapes D3-dependent inactivation in specific cellular regions. Studying thyroid hormone transport at the resolution of cellular organelles is challenging, and this work used unique approaches to achieve this goal. The results support the conclusions, however, most mechanistic experiments were performed in vitro and using a pharmacological approach, and further experiments in mice that lack MCT8 and D3 (preferably inducible system), would have strengthened the outcomes.

Comments

1. The title suggests that this is the only mechanism of action of TH in the entire brain. It is suggested to tune it down. In addition, this mechanism may be only true to the specific tested brain region.

2. The rationale of the mechanism and the link between the results is unclear. The role of D3 and MCT8 was tested. Why this specific transporter and enzyme were selected and not other T3 transporters and deiodinase? Are these specific proteins essential to the mechanism of transport (experiments in genetic KO system can clarify this point)?

3. Abstract – "However, the steps involved in T3 signaling remain poorly known given that neurons express high levels of type 3 deiodinase (D3), which in activates both T4 and T3." Not all neurons express D3. The amount and activity of D3 can be tightly regulated in the transcriptional, translational, and activity (substrate-ligand interaction) levels. The colocalization of both T3 and D3 doesn't necessarily mean inactivation. The "paradox" should be better explained. The interaction between substrate (T3) and enzyme (D3) is highly dynamic, and can be regulated in several genetic and biochemical levels.

4. T3 may diffuse to other brain areas by other mechanisms (extracellular fluids? Transport by supporting glia cells?), and not by axon transport. The controls in the mouse experiments are not satisfactory. Local genetic protein inactivation can help to answer this key question.

5. In the MCT8 immuno-electron microscopy experiments, control data will help. A similar experiment in MCT8-KO neurons can differentiate signal from noise.

6. The temporal dynamics of T3 transport is not clear. In the in vitro experiments, it takes 3 days. However, T3 response can be much faster. Gene expression can change just hours post T3 injection. In addition, the author should explain the possible discrepancy between the time it takes to monitor Luc mRNA (24 h) and axonal T3 transport (72h).

7. Why XH (D3 inhibitor) was not used in axon to soma experiments as it was used in the soma to axons experiments.

8. In general, most of the mechanistic conclusions are based on in vitro experiments. This limitation should be underlined in the abstract and conclusions.

9. Experiments with MCT8 and D3 KO cells and mice (for example, Víctor Valcárcel-Hernández et al. 2022, which showed neurological phenotype), as well as rescue experiments would have strengthened the conclusion.

---

## [Author Response]

Reviewer #1 (Recommendations for the authors):In conclusion, there are a lot of weaknesses in the conclusions of a very ambitious study. I believe that it would be better to split it into two articles. My feeling is that the connection between the two parts (deiodination and transport) is artificial, while the experiments presented in Figures 3 and 4 represent the most promising part of the article. It would be important to reinforce each part by using KO mice, or an alternative genetic strategy, to reinforce the conclusions.

Thank you. We have addressed all points raised by the reviewer and in many cases generated new data, which is now included in the new version. The genetic approach (KO mice) was considered but we were discouraged by the weak/absent phenotype of the Mct8KO mouse. We considered that figuring out what, during development, compensates for the Mct8KO inactivation would be beyond the scope of the present investigation.

Reviewer #2 (Recommendations for the authors):The main concerns raised by this reviewer were detailed in the public review. Here I make additional suggestions for potential improvements.1) Furthermore, experiments in Figure 4 seem to present an average of only two independent experiments, which are not sufficient to draw robust conclusions.

We agree and have repeated these experiments to increase the sample size; now, n = 4—figure 1 C and H.

2) It would be helpful to include a paragraph discussing the limitations of this study in the Discussion.

Thank you. A paragraph with the limitations of the study was added to the discussion.

3) Figure 2: The hippocampus is enriched in GFAP-positive astrocytes, whereas some other brain regions (e.g. the cortex) are not so (e.g. 10.1155/2019/9605265). Although not the main topic of study here, these different astrocyte phenotypes may result in different outcomes when comparing the cortex and hippocampus. Further, do hippocampal astrocytes present altered reactivity in Thr92Ala-DIO2 mice? A few sentences in the discussion could be interesting.

Thank you. We agree that a better characterization of the astrocytes isolated from different regions could be useful to understand our model. Nonetheless, these studies were removed from the manuscript.

Reviewer #3 (Recommendations for the authors):As stated in the public review, although 2 topics that they deal with in this manuscript are interesting, it may not be appropriate to deal with two completely different topics in one paper. I rather suggest deleting table one, Figure 1, and Figure 2, and re-write the paper with other data. Even after such a modification, this paper is still very attractive, although several additional experiments may be required.

Thank you. We removed from the manuscript the experiments with DIO2 polymorphism. Therefore, all comments specifically connected to this part of the manuscript will only be briefly addressed

It is interesting to examine retrograde axonal transport using a compartmentalized chamber. However, since primary cortical neurons contain a different subset of neurons, particularly excitatory (glutamatergic) and inhibitory (GABAergic) neurons, it is necessary to characterize further whether this transport is specific to a certain subset of neurons or ubiquitous to all of them.

Thank you. Reviewer 1 had a similar suggestion. To satisfy both reviewers, we have done new immunofluorescence studies and found that most of the neurons residing in the MC-CS are excitatory, exhibiting the marker vesicular glutamate transporter 1 (Vglut1). No inhibitory neurons were immunoreactive when incubated with an Ab against an isoform of the glutamate decarboxylase (GAD67). These new results have been incorporated in the manuscript (Figure 5—figure supplement 5).

In the same line, regarding the microinjection of T3 into the median eminence, several different neuroendocrine neurons are located in the periventricular nucleus such as CRH, vasopressin, oxytocin, and somatostatin neurons, in addition to TRH neurons. Thus, it is necessary to characterize further whether retrograde transport is seen in TRH neurons to prove their hypothesis.

Thank you. We know from our previous studies of the mouse median eminence that thyroid hormone signaling might not be homogeneous among all types of PVN neurons (PMID: 22719854). For example, we detected D3 in about 25% of the TRH neurons, and about 70% of the GnRH, GHRH, and CRH neurons, while no D3 was detected in SST neurons. Also, the axons from these neurons expressed abundant MCT8 protein levels. Aditionally, we found out that induction of local hyperthyroidism in the median emminence that indcuces T3-mediate downregulaton of TRH in the PVN (PMID: 36322711). While we agree with the reviewer that knowing the specifics of each neuron type would be informative, studying T3 transport in each type of neurosecretory PVN neuron projecting to the median eminence would require a specific effort to overcome the intrinsic technical challenges. We have revised our statements in the manuscript and softened our rationale/conclusions when appropriate.

Although I suggest deleting table 1, I have a major comment regarding this table. In this table, the changes in LUC mRNA were indicated by arrowheads. Thus, it is rather difficult to compare the pattern of changes between Thr92 and Ala92 mice. Relative levels of LUC mRNA should be shown.

Thank you. These studies were removed from the new manuscript version.

[Editors' note: further revisions were suggested prior to acceptance, as described below.]

Reviewer #4 (Recommendations for the authors):Salas-Lucia et al. investigated novel Triiodothyronine (T3) transport and action in neurons with an in vitro system (compartmentalized microfluid device) and an in vivo experiment (implantation of T3 crystals into the brain cortex). The current version of the manuscript was updated according to new experiments based on previous comments. Although it still shows exciting data for this field, some concerns still exist as below.1) In the previous version of the manuscripts, the authors found retrograde T3 transport in neurons (Figure 2I) as a novel finding. On the other hand, the new experiment (Figure 2-S3B) also showed the same rate of anterograde T3 transport from MC-CS to MC-AS. Also considering that MCT8 is expressed in every part of neurons, T3 is transported into neurons everywhere and by both anterograde and retrograde way in the same rate. So, T3 in neurons is homogenously distributed and circulated by bidirectional transport through microtubules. According to those results, I agree that the retrograde transport T3 in neurons is a novel finding, but I feel that the impact of this finding is not striking.

Thank you. The present study specifically demonstrates for the first time that T3 is a cargo to the endosomes. These non-degradative endosomes act as “trojan horses” to protect T3 from D3-mediated metabolism. This is an absolutely novel finding with biological relevance, as the axonal traffic of T3 can regulate T3 signaling and gene expression in neurons. Until now, we had no idea how T3 bypassed the relatively high D3 activity in the neurons and could regulate gene expression. The present investigation delineates those mechanisms.

2) In the first part of the experiment, [1]T3125 is added to MA-AS, [2] T3125 is transported into neurons through MCT8 and transported to cytosol. [3] some T3125 is converted T2 by Dio3 and some other T3125 is transported again (by MCT8) and appeared in MC-CS. Although the authors could detect the T3125 in such as Figure 2 IJ(orange panel) and Figure 2-SBC(blue panel), but its levels are very low. This data raises doubts about accuracy. So I would suggest that how the concentration of T3125 in neuron without medium at MC-CS. MCT8 locating in MC-CS may significantly limit the transport of T3125 from neuron to medium.

Thank you. The UPLC readings clearly show a distinct peak of T3 in the MC-CS, which is significantly different from the background. This is a reproducible finding. We already did what the reviewer asked by measuring 125IT3 in the nuclei of the neurons (pls see Figure 2E ).

3) In Figure2-S2C, the authors confirmed the results in Figure2 by using rat DRG neurons. The order of Y-axis is completely different from Figure 2. How do the authors interpret it?(Also, Figure 2-S2 should be described in a similar fashion to Figure 2. Please describe the meaning of the asterisk, too.)

Thank you. We have now prepared a new graph comparable to the one presented in Figure 1K. The meaning of the asterisks has been clarified in the figure legend and now reads: “*P <0.05 when compared T3 vs. T3+SC and vs. T3+CO.”

4) In the second part, the authors implanted T3 crystals or T3 labeled by radioactive iodine to prove the retrograde transport of T3 in rat brain. As reviewer 1 pointed out in the previous comments, it is difficult to interpret. Because there is no direct evidence of retrograde T3 transport in Figure 5. Although the authors showed no T3 effects on both ipsi- and contr- lateral hypothalamus (Figure 5K), these are still conclusions by exclusion.

Thank you. Direct evidence of retrograde T3 transport is provided in the hypothalamus experiment. Here 125T3 was found in the hypothalamus just 20 min after being injected in the ME. Only background levels were detected in immediately adjacent regions. We understand your wish to see similar evidence in the experiment involving cortical implantation of the T3 crystal, but having provided direct evidence in the hypothalamus experiment, here we opted for showing that the retrogradely transported T3 could actually change gene expression. These decisions were based on the fact that these are extremely labor-intensive but complementary experiments.

Reviewer #5 (Recommendations for the authors):This work used in vitro and in vivo approaches to understand thyroid hormone signaling at cellular resolution. The results show that T3 is transported by the thyroid hormone transporter MCT8 into neuronal axons. In the neurons, it is transported to the cell nucleus and activates gene expression. Type 3 deiodinase (D3) inactivates T3 signaling in specific cellular localization. These findings provide a cellular mechanism for T3 transport and activity, which is supported by experiments in both compartmentalized microfluid device and mouse systems.The goal of this work is to study thyroid hormone (T3) transport and signaling at cellular resolution. Salas‐Lucia and colleagues used two models – a microfluid device and mice. They showed that T3 enters into neuronal axons and transport by non-degradative lysosomes (NDLs) to the nucleus, where it activates thyroid responding genes. This process is regulated by the thyroid transporter MCT8 and type 3 deiodinase (D3). In addition, the findings explain how T3 escapes D3-dependent inactivation in specific cellular regions. Studying thyroid hormone transport at the resolution of cellular organelles is challenging, and this work used unique approaches to achieve this goal. The results support the conclusions, however, most mechanistic experiments were performed in vitro and using a pharmacological approach, and further experiments in mice that lack MCT8 and D3 (preferably inducible system), would have strengthened the outcomes.

Thank you. Both Silychristin and Xantohumol are highly selective inhibitors of MCT8 transport and deiodinase, respectively [refs 30 and 31]. Their advantage over a genetic approach is that they can be (and were) used on a specific side of the microfluid compartment, testing the roles of MCT8 and D3 on specific portions of the neurons. This would not be possible with a genetic approach. This has been explained extensively during the previous round of reviews.

Comments1. The title suggests that this is the only mechanism of action of TH in the entire brain. It is suggested to tune it down. In addition, this mechanism may be only true to the specific tested brain region.

Thank you. The title has been modified. The new title reads: “Axonal T3 Uptake and Transport Can Trigger Thyroid Hormone Signaling in the Brain”

2. The rationale of the mechanism and the link between the results is unclear. The role of D3 and MCT8 was tested. Why this specific transporter and enzyme were selected and not other T3 transporters and deiodinase? Are these specific proteins essential to the mechanism of transport (experiments in genetic KO system can clarify this point)?

Thank you. It is well known that D3 is the only deiodinase expressed in neurons. It inactivates thyroid hormone. Thus, it is logical to try and understand its role. As explained in the introduction, it was not known how T3 could bypass D3 and reach the nucleus to affect gene transcription. MCT8 is the T3 transporter that, when mutated, causes the devastating syndrome AHDS. Mutations in other transporters do not cause this. This is explained in the introduction as well.

3. Abstract – "However, the steps involved in T3 signaling remain poorly known given that neurons express high levels of type 3 deiodinase (D3), which in activates both T4 and T3." Not all neurons express D3. The amount and activity of D3 can be tightly regulated in the transcriptional, translational, and activity (substrate-ligand interaction) levels. The colocalization of both T3 and D3 doesn't necessarily mean inactivation. The "paradox" should be better explained. The interaction between substrate (T3) and enzyme (D3) is highly dynamic, and can be regulated in several genetic and biochemical levels.

Thank you. We wonder what the evidence is that not all neurons express D3. We agree that some neurons may express more and others less, but we are unaware that some sets of neurons might not express D3 at all. The presence of D3 is clear in all neurons that we studied. Dio3 is regulated transcriptionally. We have shown that the D3 subcellular localization is affected by hypoxia. We are aware of one paper that suggests that D3 can be regulated by its own substrate, T3. To our knowledge, results in this paper were never reproduced, certainly not in our lab. Not sure we understand what the colocalization between D3 and T3 is the reviewer is referring to. The paradox is that neurons express high D3 levels and yet do not metabolize T3 in their axons. Just look at the catabolism of T3 when the tracer is added to the cell side of the microfluid chambers.

4. T3 may diffuse to other brain areas by other mechanisms (extracellular fluids? Transport by supporting glia cells?), and not by axon transport. The controls in the mouse experiments are not satisfactory. Local genetic protein inactivation can help to answer this key question.

Thank you. We respectfully disagree with the reviewer. In the experiments with the hypothalamus, the control areas were just 1-2 millimeters away from the PVN. Any glial-based or extracellular fluid-based diffusion, as suggested by the reviewer, would have affected the results here, but the controls were negative.

In the experiments with the cortex, please recall that the hypothalamus (negative control) is located at roughly the same distance from the area where the T3 crystal was implanted as the cortical site on the other hemisphere. In this case, a glial-based or extracellular fluid-based diffusion would have affected the hypothalamus, but so sign of T3 stimulation was observed.

We are unsure about the suggestion posed by the reviewer. Using “local genetic protein inactivation” would not have been able to isolate different parts of the neurons (as we did in our experiments) or avoid T3 diffusion as suggested by the reviewer. Also, we are wondering how astrocytes could play a role in the interhemispheric transport of T3. Since it is well known that astrocytes are not known to project across long distances, we have not considered this possibility.

5. In the MCT8 immuno-electron microscopy experiments, control data will help. A similar experiment in MCT8-KO neurons can differentiate signal from noise.

Thank you. These experiments with MCT8-KO neurons were done previously in our laboratory. The MCT8 signal obtained in the present experiments is identical to the ones observed in the previous publication. We refer to the reviewer to Figures7 and 8 of our previous work–PMID: 22719854.

6. The temporal dynamics of T3 transport is not clear. In the in vitro experiments, it takes 3 days. However, T3 response can be much faster. Gene expression can change just hours post T3 injection. In addition, the author should explain the possible discrepancy between the time it takes to monitor Luc mRNA (24 h) and axonal T3 transport (72h).

Thank you. Please consider that the two approaches used to demonstrate neuronal T3 transport in vivo are fundamentally different. In the in vitro experiments, we were able to measure T3-induced gene expression after 24h. However, the detection limit of our UPLC-γ counter did not allow us to detect 125I-T3 transport earlier than 72h (this is because 125I-T3 accumulates on the other side of the compartment). This is the explanation for the “discrepancy” as pointed out by the reviewer.

7. Why XH (D3 inhibitor) was not used in axon to soma experiments as it was used in the soma to axons experiments.

Thank you. We respectfully direct the reviewer’s attention to fig2K of the previous version, now fig1K, and to lines 350 to 351, where they will find the suggested experiment.

8. In general, most of the mechanistic conclusions are based on in vitro experiments. This limitation should be underlined in the abstract and conclusions.

Thank you. We have included a new line in the limitations that stresses the need for more studies using in vivo models–lines x to x.

9. Experiments with MCT8 and D3 KO cells and mice (for example, Víctor Valcárcel-Hernández et al. 2022, which showed neurological phenotype), as well as rescue experiments would have strengthened the conclusion.

Thank you. We respectfully refer the reviewer to the answer to their first question. A genetic approach would have allowed us to answer the questions about the role of D3 and MCT8 in T3 uptake and transport in neurons. It would not have allowed us to study the role of MCT8 and D3 on each side of the microfluidic compartment.